# Predicting neuronal firing from calcium imaging using a control theoretic approach

**Nicholas A. Rondoni** [ID][1]*, **Fan Lu**[1], **Daniel B. Turner-Evans** [ID][2], **Marcella Gomez**[1]

**1** Department of Applied Mathematics, University of California Santa Cruz, Santa Cruz, California, United States of America, **2** Department of Molecular, Cell and Developmental Biology, University of California Santa Cruz, Santa Cruz, California, United States of America

* nrondoni@ucsc.edu

**Data availability statement:** All Python source code and outputs are available in the following repository: https://github.com/N-Rondoni/slugFind.

## Abstract

Calcium imaging techniques, such as two-photon imaging, have become a powerful tool to explore the functions of neurons and the connectivity of their circuitry. Frequently, fluorescent calcium indicators are taken as a direct measure of neuronal activity. These indicators, however, are slow relative to behavior, obscuring functional relationships between an animal's movements and the true neuronal activity. As a consequence, the firing rate of a neuron is a more meaningful metric. Converting calcium imaging data to the firing of a neuron is nontrivial. Most state-of-the-art methods depend largely on non-mechanistic modeling frameworks such as neural networks, which do not illuminate the underlying chemical exchanges within the neuron, require significant data to be trained on, and cannot be implemented in real-time. Leveraging modeling frameworks from chemical reaction networks (CRN) coupled with a control theoretic approach, a new algorithm is presented leveraging a fully deterministic ordinary differential equation (ODE) model. This framework utilizes model predictive control (MPC) to challenge state-of-the-art correlation scores while retaining interpretability. Furthermore, these computations can be done in real time, thus, enabling online experimentation informed by neuronal firing rates. To demonstrate the use cases of this architecture, it is tested on ground truth datasets courtesy of the *spikefinder* challenge. Finally, we propose potential applications of the model for guiding experimental design.

## Author summary

We put forward a novel approach to infer when neurons fire as a function of calcium concentration. These calcium recordings are useful for imaging whole populations of neurons, such as those found in the brain, but act only as a proxy for the true underlying spiking occurrences. Moreover, these calcium traces are unfortunately noisy. To uncover the actual firing times we apply a control theoretic approach to a model derived from chemical reaction equations. The result is competitive with the state of the art, fast enough to provide information in real time, and highly interpretable. More broadly

**Funding:** This effort was supported by the SciAI Center, and funded by the Office of Naval Research (ONR), under Grant Number N00014-23-1-2729 to MG. The funding agency played no role in study design, data analysis, decision to publish, or preparation of the manuscript.

**Competing interests:** The authors have declared that no competing interests exist.

this analysis process aids understanding in contexts where methods of measurement obfuscate the desired ground truth information. To demonstrate potential applications of the model we quantify how biochemical properties of the indicators, which allow the tracking of calcium, impact prediction accuracies. In more general terms, this framework has the potential to enable understanding of the tools used to measure the desired underlying signals.

## Introduction

Information processing within populations of neurons is frequently uncovered via calcium imaging [1]. This method indirectly tracks concentrations of intracellular calcium ions by their fluorescence, which is a noisy realization of the underlying calcium signal [2]. An action potential within a neuron results in an uptake in intracellular calcium, in turn yielding an increase of fluorescence as calcium ions bind to an indicator. Downstream analyses are frequently concerned with true spiking times, not fluorescent traces. Converting between a fluorescent trace and true spiking times is nontrivial, coming with many computational difficulties. Amongst them are disparities in the temporal resolution of imaging compared to neuronal dynamics, noise from the recorded fluorescent signals, nonlinear relationships between calcium indicators and fluorescence, and biological parameters unknown a priori [3].

This work is concerned with inferring a true firing signal from noisy time series calcium imaging data. State of the art methods approach this in a handful of different ways. Ranging from supervised learning [4], generative methods [5], sophisticated particle filtering [3], and nonnegative deconvolutions manifesting as convex optimization problems [6,7], a litany of theory has arisen as a means to uncover the true spiking occurrences of a neuron. The works of [4,8] and later [9] present thoughtful surveys of leading methods, while suggesting their preferred approach.

Of particular interest as of late is the ability to analyze systems of neurons in real time. Closed loop investigation of neural circuitry requires quick data processing and carefully crafted experimental conditions. With these conditions met, experiments can deliver sensory stimuli informed by system-wide neural dynamics. For example, a closed loop strategy was employed to tease out how activity in visual processing regions are connected to specific brain states [10]. Recent work from [11] has shown how real time inference of neural activity can inform what stimulation should be supplied to the neurons undergoing imaging, though these authors use calcium traces directly as their measures of activity. Model predictive control (MPC), tracing back to 1960s [12], has caught attention in biological control systems, resulting in high levels of control accuracy [13,14]. This controller leverages a model, often based on ordinary differential equations, and is used to predict optimal control strategy via optimization over a finite time window of data. Neuronal firing is treated as a control input into the system of equations. The MPC algorithm computes this neuronal firing signal with the goal of tracking measured calcium traces with the model. The performance of our algorithm is compared to two state-of-the-art methods. With only a handful of methods fast enough to be considered real time [7], this work adds to this body of existing methods.

## Methods and models

This section utilizes a chemical reaction network to derive a model, then discusses how MPC can be tailored to infer underlying firing rates.

## Chemical reaction formulation

Starting with first principles, we examine the chemical reaction network theorized to dictate the interplay between a calcium ion, a calcium indicator, and a bound (or fluoresced) calcium indicator. These quantities will be denoted $[Ca^{2+}] = x(t)$, $[CI] = y(t)$, and $[CI^*] = z(t)$ respectively, where $[\cdot]$ denotes concentration. Intracellular calcium binds to an indicator at a rate of $k_f$. Simultaneously these compounds may unbind at a rate of $k_r$, leaving the indicator and calcium ion free within the cell once again. This gives

$$Ca^{2+} + CI \; \underset{k_r}{\overset{k_f}{\rightleftharpoons}} \; CI^* \tag{1}$$

With this chemical reaction network, we may utilize the law of mass action, which posits the rate of a reaction is proportional to the product of reactant concentrations [15,16].

Supposing calcium indicators do not passively diffuse out of the cell, the total concentration of calcium indicator is time invariant. That is the sum of concentrations of the indicator, both bound and unbound, should remain constant. Call this value $L$. Then for all time $t$

$$y(t) + z(t) = L \qquad L \in \mathbb{R}^+$$

and the equations that follow from (1) may be reduced to

$$\begin{cases} \dot{x} & = k_r z - k_f x(L - z) \\ \dot{z} & = k_f x(L - z) - k_r z \end{cases} \tag{2}$$

Note these ODEs are nonlinear due to the product of $x$ and $z$ appearing in both equations. Motivated by the observation that a neuron's firing results in an uptake of calcium [17], and ultimately is the dominating force controlling the balance of $x = [Ca^{2+}]$, we add in a continuous function of time $s(t)$ to represent firing rate at a given time $t$, scaled by some constant $\alpha$. We finally note calcium ion $x$ could passively diffuse out of the cell, and subtract a $\gamma x$ term to account for this, arriving at the final governing ODE system

$$\begin{cases} \dot{x} & = \alpha s - \gamma x + k_r z - k_f x(L - z) \\ \dot{z} & = k_f x(L - z) - k_r z \end{cases} \tag{3}$$

where

$$x(t), z(t) : \mathbb{R}^+ \cup \{0\} \to \mathbb{R}, \qquad \alpha, \gamma, k_r, k_f, L \in \mathbb{R}^+$$

In this phrasing, the question of inferring spiking rates as a function of calcium imaging has been cast as an inverse problem, in which we learn the control signal responsible for driving the observed dynamics. A table of parameter values and their biophysical interpretation can be found in S1 Table. Justification that this system is stable for all achievable values of $s$ is in S1 Appendix. An auto-calibration pipeline, which infers reasonable parameter value(s), is outlined in S2 Appendix. Parameters of the model are optimized on 1-3 cells for each dataset, using the calcium signal and ground truth data obtained via a patch clamp method.

## Model predictive control and the neuron

MPC is an optimal control technique. This well tested method computes necessary control actions that minimize a cost function and adhere to an underlying model [18], in our case a constrained set of ODEs defined in Eq (3). This control action is computed over a finite horizon, enabling changes to actuation that update with real time information. For all simulations in the results section, the open-source MPC package do-mpc [19], implemented in python, was utilized.

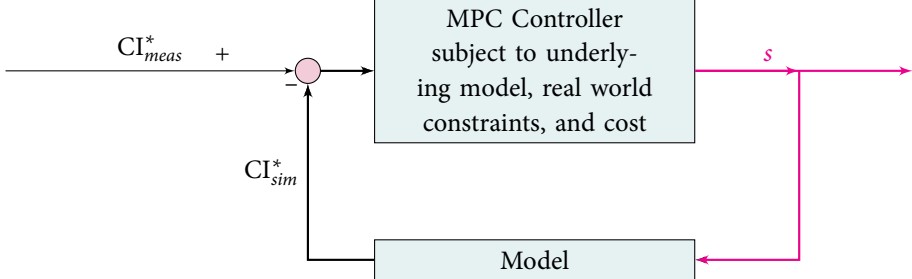

In the above block diagram, $CI^*_{meas}$ denotes the measured amount of fluoresced indicator, while $CI^*_{sim}$ represents the simulated amount of this quantity as dictated by the model (3). Per the definition of MPC, we require a cost defined over our horizon of $n$ timesteps. For simulations to follow, we consider the following minimization problem

$$\min_{s\in\mathbb{R}^+}\sum_{k=0}^{n-1}\left(\left(CI^*_{k,sim}-CI^*_{k,meas}\right)^2+r\left(s_k-s_{k-1}\right)^2\right)+\left(CI^*_{[0,T],sim}-CI^*_{[0,T],meas}\right)^2 \qquad (4)$$

for parameter $r\in\mathbb{R}^+$ and current simulation time $T$. The first squared term enforces our simulated and measured calcium indicators track one another on a finite horizon, and $s_k-s_{k-1}$ penalizes changes in $s$, essentially limiting the derivative and encouraging smoothness. All simulations to follow take $r = 0.01$ as regularization term. Other authors have had success with the similar cost functions, sometimes using a $l_0$ or $l_1$ penalty on $s$ instead of the quadratic cost above [6,7]. Observe that the final term in (4) acts as terminal penalty function, which aids the effectiveness of MPC. An improperly chosen terminal penalty function may degrade performance and potentially destabilize the closed-loop system. Although increasing the prediction horizon can improve performance, this significantly raises computational costs. A well-established solution to these issues involves selecting the terminal penalty function as the infinite-horizon value function that satisfies the dynamic programming equations [20,21], as is done above. The horizon length $n = 6$ was utilized for all simulations corresponding to a 60 ms time window.

In order to partially address the noisy fluorescence signal, a shifted sigmoidal filter

$$\sigma(x) = \frac{1}{1+e^{-(x+1)}} - h$$

is applied to $CI^*_{meas}$. To avoid saturation of this filter, e.g., $CI^*_{sim}$ values becoming stuck at 1, a vertical shift downward of $h = 0.15$ is additionally used. If saturation still occurs, a more aggressive vertical shift of $h = 0.25$ is then applied and computations are restarted. The nature of a sigmoidal curve minimizes contributions from extreme values, mapping them closer to 0 or 1, while retaining a linear regime for middle of the pack measurements. More advanced

filtering techniques to address stochasticity inherent in the system could undoubtedly improve this methodology, this is discussed briefly in the following section.

## Results and discussion

With a model formulated and the tools of inference defined, this section benchmarks performance using the *spikefinder* [22] challenge dataset. This dataset is comprised of recordings from two diferent benchmarking efforts, that of [4] and another from the cai-1 CRCNS website [23]. Our software implementation is publicly available (code: https://github.com/N-Rondoni/slugFind).

### *Spikefinder* validation

This dataset contains not only time series calcium imaging data, but recorded "ground-truth" spikes via electrophysiology. It is one of the few public datasets to contain simultaneous recordings of spiking times and fluorescent traces, allowing for validation of many of the models discussed in this document. An application of our methodology, henceforth called the MPC approach, to a particular neuron produces a time series of simulated bound calcium indicator $CI^*_{sim}$ and spiking rates $s$ as a function of measured indicator $CI^*_{meas}$, visualized below in Fig 1.

The primary metric valued during the *spikefinder* challenge was Pearson correlation coefficient. While emphasis is still placed on correlation, alternative methods of measuring spike synchrony have been put forward such as Victor-Purpura and van Rossum metrics [24–26]. Since spikes are inherently discrete events, for comparison of two spike trains to be computationally tractable and informative we downsample in accordance with the *spikefinder* challenge methodology. Specifically, measured and simulated spiking signals are downsampled from their native acquisition rate of 10 ms by a factor of 4, yielding a bin width of 40 ms.

Beginning with correlation coefficient, we compare against the state of the art algorithms Spike Triggered Mixture (STM) [4] and Oasis [7] in Fig 2. Here STM serves as a representative of supervised methods, while Oasis is a non-negative deconvolution or NND algorithm. Note the work of [7] is another method that is certainly fast enough for real time applications, alongside the MPC approach put forward here. Oasis was eventually benchmarked on the spikefinder dataset by [9].

As can be seen in Fig 2, our mean correlation coefficient has a .01 difference relative to STM. The difference in mean with respect to Oasis is larger; about 0.167. To explore this on a more granular level, we present a neuron by neuron comparison of the correlation coefficients produced by both STM and Oasis versus MPC in Fig 3. It's interesting to note the difference in performance of Oasis when no indicator information is provided, highlighting the importance of prior information and training. Thus, we would like to caution that direct comparison across methods is difficult to ascertain due to differences in data used for training, the handling of missing data points, and the inherent properties of a dataset itself. We highlight how performance varies across datasets in Figs 4 and 5. While it's clear that Oasis maintains a lead with respect to accuracy we would like to note that our method is the only current method that can be implemented for real-time feedback with no variable backtracking. We dedicate further discussion on this later in manuscript. We also highlight the minimal data required for training below.

The MPC algorithm does not require training in the same way a supervised approach such as STM would. Rather, parameters may be selected a priori from known chemical reaction constants, however this undoubtedly results in a sub-optimal parameter regime as $\alpha$ and $\gamma$

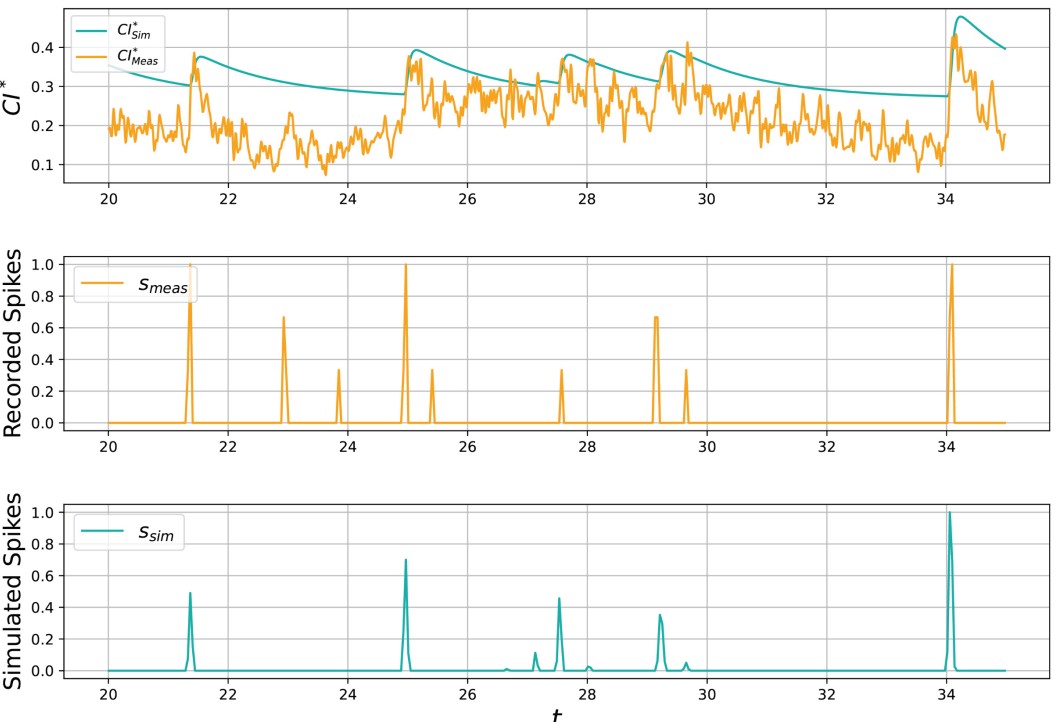

**Fig 1. Visualization of approximations** Pearson correlation coefficient is 0.694 for this particular 15 second subset of test data set 1, neuron 0. The whole 12-minute recording scores a 0.491. Spiking rates plotted above are computed by downsampling neuronal spikes and normalizing them to be within [0, 1] for ease of viewing. Some false activity is predicted, while other activity is missed. Parameter values used in this simulation can be found in table A in S1 Table.

are likely unknown. For the simulations shown in this section parameters were selected for their approximate biophysical relevance, then the auto-calibration process outlined in appendix S2 Appendix was used to infer reasonable values for $\alpha$. Values of learned $\alpha$ used in simulations by dataset are summarized in Table B in S1 Table. The same $k_r, k_f, \gamma$ were used across all datasets thanks to the auto-calibration's ability to vary $\alpha$ to accommodate as needed. Thus there is a small training phase, though much shorter than a neural net's - a single neuron recording was enough to calibrate successfully on 8 of the 10 datasets. For the remaining two, datasets 8 and 9, we required 3 neurons. Recording lengths vary from approximately 5-20 minutes each. Datasets 1-5 had train and test data available, and only a single neuron from the train dataset was used to calibrate for these particular recordings. For datasets 6-10, we used the first neuron's recording to train then tested on the remaining. In contrast to neural nets whose correlation scores are significantly higher for the neurons on which they trained, our methodology avoids such overfitting - with test and train scores not differing in a notable way.

It is challenging to find a metric that accurately captures the performance of an algorithm. Thus, we consider an alternative metric known as the Victor-Purpura distance. Note that lower scores are better when examining the Victor-Purpura distance [24]. In Fig 6 we compare the naive Oasis, meaning Oasis supplied with no information, Oasis supplied with indicator information, and STM to the MPC approach. In Fig 7 we compare neuron by

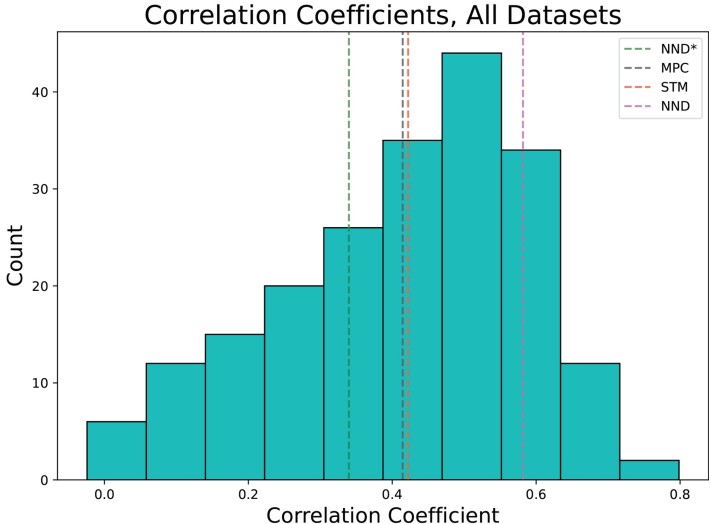

**Fig 2. Comparison to state of the art.** Histogram of correlation coefficients across all data sets, both train and test. The red dashed line denotes the mean of the STM algorithm, green denotes Oasis supplied with no information about the indicator, pink denotes Oasis with information about the indicator, and the blacked dashed line denotes the mean of the plotted dataset. Means presented are across all data sets, train and test, as well. Resultant firing signals downsampled by a factor of 4 in accordance with the *spikefinder* challenge, yielding a bin width of 40 ms.

## Comparisons with State of the Art

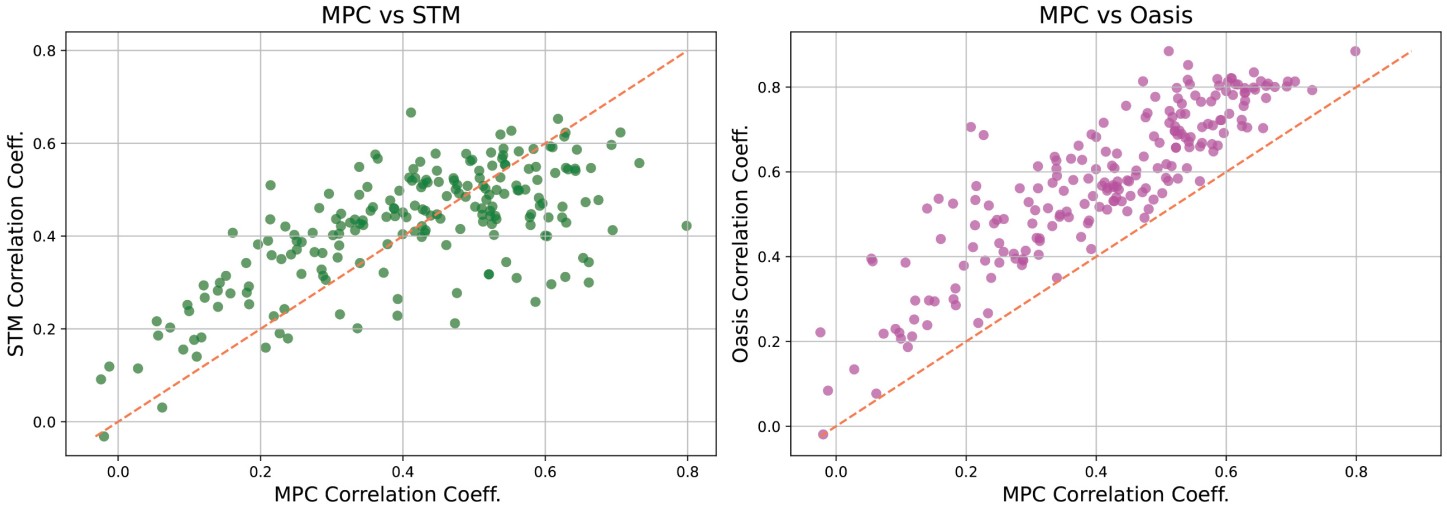

**Fig 3. Neuron by neuron comparison of correlation coefficient for the state-of-the-art methods STM and Oasis.** A dot below the line means MPC's prediction is more highly correlated, while a dot above means the other method's prediction is superior. Advantages of our method for real-time applications is discussed further in the section Remarks on real time control.

neuron results of MPC against STM and Oasis under this metric. By this metric our algorithm performs better than Oasis for some neurons. The cost of moving a spike a single timestep is 1 for this simulation.

Finally, we compare the variance in correlation score by dataset. This is summarized in the following Table 1, in which the standard deviation $\sigma_i$ is presented for dataset $i$, by method. $\sigma_{all}$

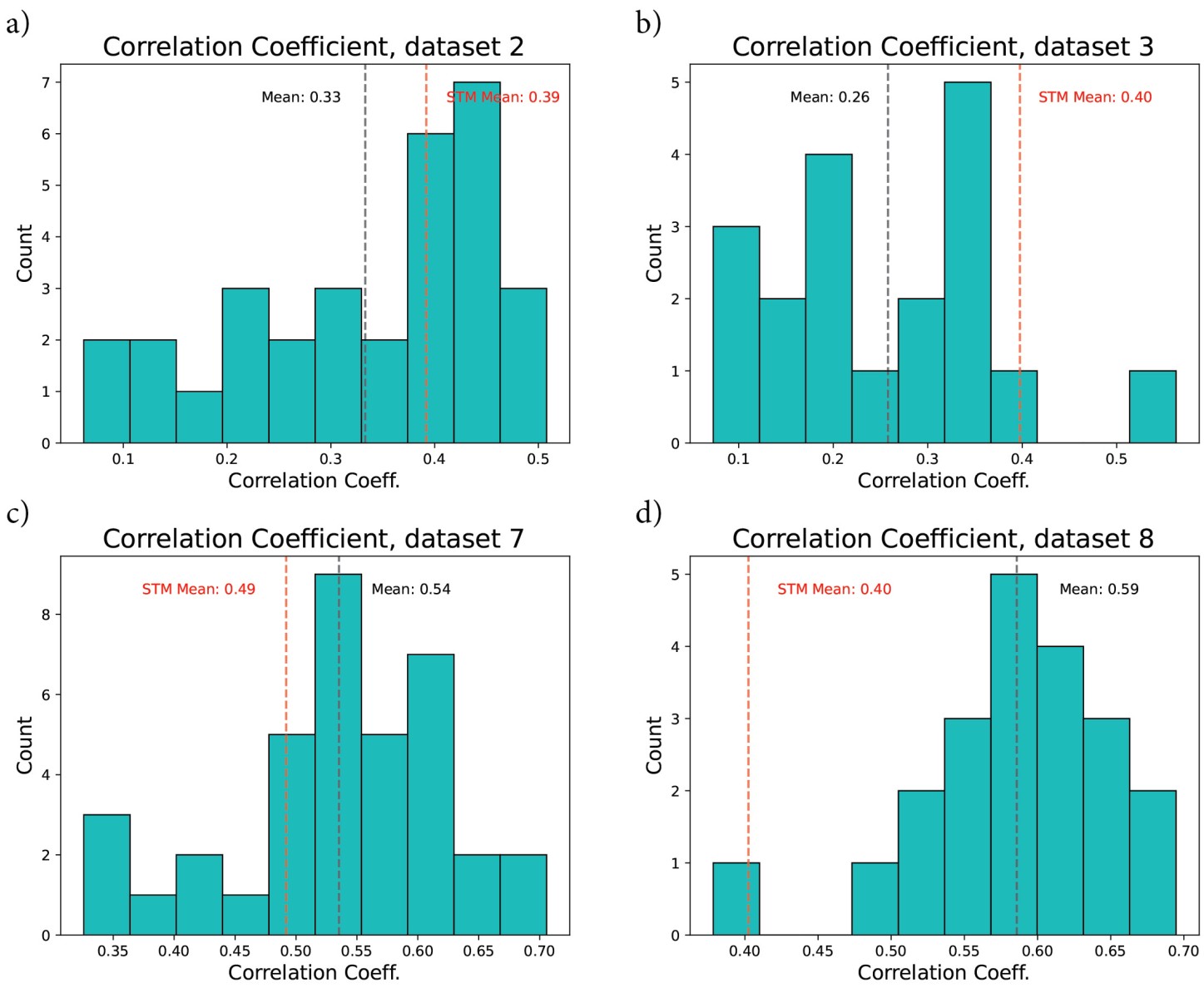

**Fig 4. Comparison to STM on specific datasets.** Histogram of correlation coefficients computed for four different datasets. a) and b) both showcase STM outperforming the MPC approach, while c) and d) show the MPC approach outperforming. Datasets 2 and 3 had both train and test data available - both are included. For datasets 7 and 8 only training data was made available.

denotes the standard deviation of all datasets together. The variance is comparable across all algorithms.

## Reflections on noise

Though this MPC approach does not require training in a conventional neural net sense, careful selection of parameters is still imperative. This can be seen by considering Figs 8 and 9 in tandem.

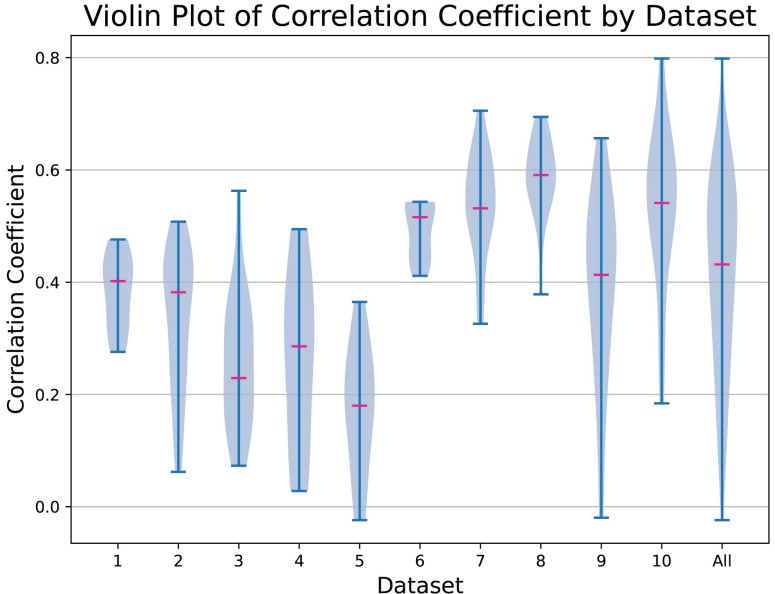

**Fig 5. Violin plot of correlation score**  Pearson correlation coefficients from MPC approach organized by data set, 40 ms bin sizes. The violin plot uses kernel density estimation to compute an empirical distribution of the samples by dataset. Medians of MPC approach are included in pink.

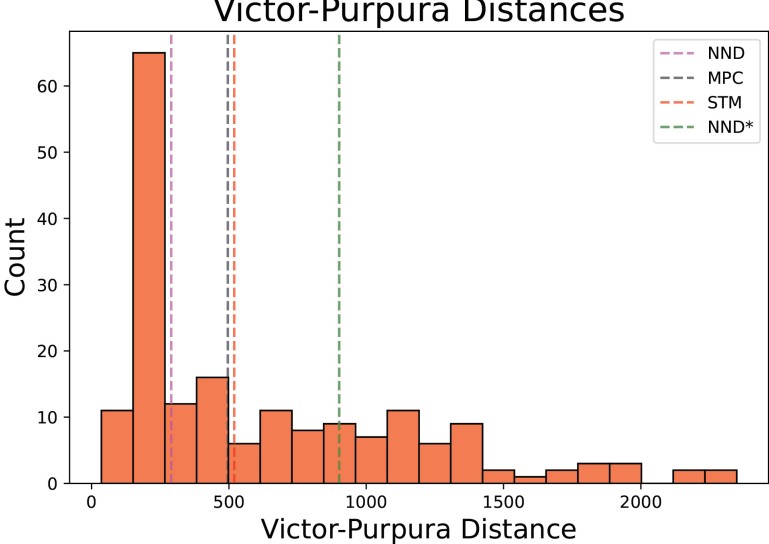

**Fig 6. Comparison of Victor-Purpura distance**  Victor-Purpura distances across all datasets. Median value of MPC approach shown with a black dashed line, while the median of Oasis supplied with no information (NND*) is shown in green. The median of Oasis (NND) supplied with indicator information is in pink, and STM is in red. Since lower is better, MPC outperforms both the naive oasis method and STM, while Oasis outperforms MPC. 20 scattered outliers of Victor-Purpura distances greater than 3500 have been excluded from the plot for ease of viewing, leaving 186 different samples visualized in this histogram. Medians were computed with all samples included.

## Victor-Purpura Comparisons with State of the Art

**Fig 7. Comparison of Victor-Purpura distance** Neuron by neuron comparison of Victor-Purpura distance for all available data. 20 scattered outliers of Victor-Purpura distances greater than 3500 have been excluded from the plot for ease of viewing, leaving 186 different samples visualized in this comparison. A lower Victor-Purpura distance is better, meaning MPC's prediction is favorable with respect to this metric when the dot is above the dashed line.

**Table 1. Comparison of method's standard deviations.** Across all datasets MPC has a smaller standard deviation than Oasis, but larger than that of STM.

|  | MPC | Oasis | STM |
|---|---|---|---|
| $\sigma_1$ | 0.064 | 0.069 | 0.065 |
| $\sigma_2$ | 0.123 | 0.134 | 0.120 |
| $\sigma_3$ | 0.118 | 0.143 | 0.112 |
| $\sigma_4$ | 0.149 | 0.181 | 0.170 |
| $\sigma_5$ | 0.109 | 0.129 | 0.125 |
| $\sigma_6$ | 0.050 | 0.055 | 0.083 |
| $\sigma_7$ | 0.092 | 0.079 | 0.079 |
| $\sigma_8$ | 0.071 | 0.061 | 0.067 |
| $\sigma_9$ | 0.163 | 0.176 | 0.138 |
| $\sigma_{10}$ | 0.137 | 0.161 | 0.135 |
| $\sigma_{all}$ | 0.173 | 0.178 | 0.127 |

In Fig 8 near optimal parameters are utilized, which incorporates a slow decay rate. This slow decay rate filters out noise in the fluorescence signal by avoiding small increases not worthy of a spike. In contrast, Fig 9 has parameters selected to encourage almost perfect tracking of $CI_{meas}$. While this is easily accomplished via the MPC algorithm, the resultant signal is riddled with false spikes, and thus has a low correlation coefficient. This is because under these circumstances the algorithm tracks the *noise in imaging* instead of the true underlying calcium signal. As mentioned in the introduction, fluorescence is a noisy filtered realization of the true calcium trace, and as such the MPC approach would certainly be improved by incorporating a nonlinear conversion between normalized fluorescence and $CI^*_{meas}$. Authors of MLspike [5] incorporate such ideas to great success. In contrast our current formulation equates these two quantities.

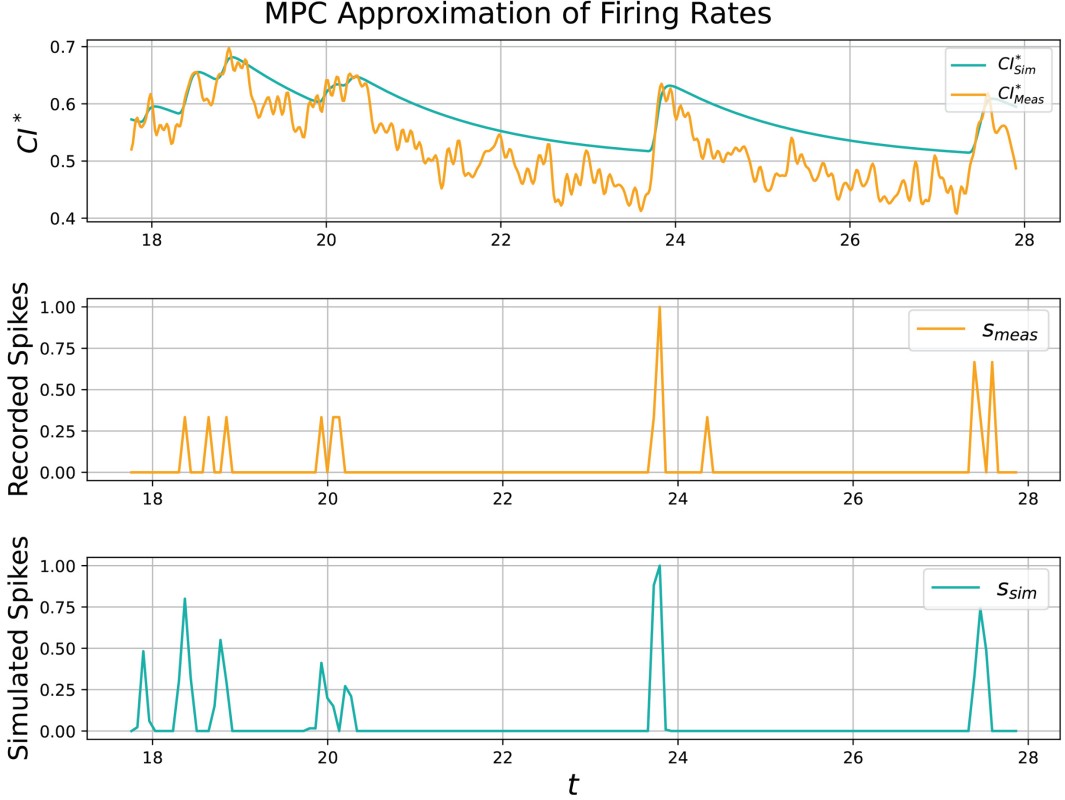

**Fig 8. Pearson correlation coefficient for this specific 10 second subset is 0.61, though the whole 12 min recording scores a 0.50.** Spiking rates plotted above are computed by downsampling neuronal spikes and normalizing them to be within [0, 1] for ease of viewing. Though most activity is detected, some is missed at around 24s.

### Remarks on real time control

Regarding the usability of methods presented in this document for online control, when training is accounted for, only our MPC approach and the NND algorithms are fast enough or structurally able to complete in real time. That is, for all solutions presented, the MPC algorithm was able to predict spiking signals in equivalent or less time than the duration of the recording. Though these signals were computed offline, the methodology needs no adaption to run out of the box in real time. To the best of our knowledge, the algorithm presented in this manuscript is the first one proposed that can be used for real-time feedback control with no variable backtracking required.

The methodology of Oasis and the scores presented from this method were computed with "backtracking" permitted. That is, predictions depended on looking back some $\Delta t$ timesteps to at most the last spike, then optimizing over this window if constraints are violated. In the original Oasis papers the authors explore the effects of this lag on prediction, and note that while 5-7 frames of lag behave similarly, there is an impact if backtracking is limited to 1-2 frames. While the algorithm is still impressively quick, if spikes are infrequent occurrences, this look back duration could be rather long. Our method requires a look ahead of six steps, which amounts to an effective lag of 60ms. Further studies are needed to understand the tradeoff between the two methods. In an application where feedback control is used to drive the cells towards a desired spiking pattern, the reference signal can be generated (replace

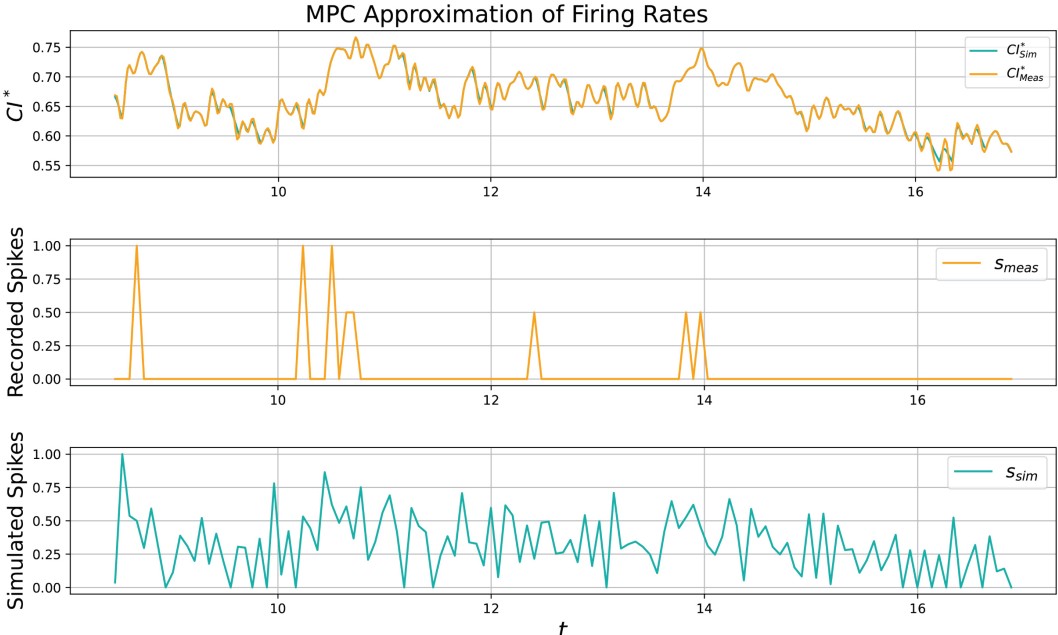

**Fig 9. In this example the MPC algorithm perfectly tracks the noisy signal CI\*$_{meas}$, which results in many false spikes.**
The correlation coefficient is computed to be 0.19. Spiking rates plotted above are computed by downsampling neuronal spikes and normalizing them to be within [0, 1] for ease of viewing.

$CI^*_{meas}$ with $CI^*_{desired}$) and the MPC algorithm will drive neuronal stimulation to achieve the desired behavior. In this framework our method works out of the box with no lag.

Furthermore, MPC is uniquely designed to handle discrepancies in measured and predicted values, through its receding horizon approach, where the control policy is recomputed at each timestep based on updated measurements. This continual re-optimization ensures that transient deviations do not necessarily lead to instability in real-time applications. Previous studies [27] have demonstrated that MPC remains relatively robust against model imperfections and measurement noise, provided the system is appropriately tuned.

While Jewell's updated methodology (along with other efficient NND approaches) can impressively compute spiking times for data sets with as many as 100,000 time instants in mere seconds on a laptop, this first requires knowledge of a parameter or training to be accurate. In Jewell's approach the relevant parameter was discovered by training on the first half of a data set then testing on the second half, suggesting training is still important if high accuracy is desirable. It is worth noting the authors have a way to infer a reasonable parameter regime, though this was not used for testing. In the case of the suite2p implementation of Oasis [9,28], training is not required but knowledge of the system must be supplied in some way to bring scores in line with means presented in plot 2. Otherwise, the means will be closer to the naive Oasis* instead of the more accurate Oasis.

The MPC algorithm takes longer than these NND methods; with run times always 30-60% less than the duration of their recording when computed on a laptop with an Apple M3 processor. However there is a rich legacy of MPC used for real time control, with many packages optimized for speed and ease of implementation, and as such is relevant in this context for its accuracy and tractable nature. Moreover, this MPC approach allows for other quantities

of interest to be inferred, such as the concentration of unbound calcium indicator within the cell. This is a result of our model's rooting in biophysical mechanisms.

## A discussion on biophysical insights

With the MPC approach and the model underlying it justified, analysis of the system may be considered. The following serves as a discussion on the potential interpretability of the model, though more experimentation is required for validation. Thus far we have been told the status of state variables and inferred $s$ from this signal. Reaping the benefits of a fully mechanistic model, we may now suppose $s$ and analyze the impacts on state variables.

To accomplish this, we perform frequency response analysis on the system of ODEs (3),

$$\begin{cases} \dot{x} & = \alpha s - \gamma x + k_r z - k_f x(L - z) \\ \dot{z} & = k_f x(L - z) - k_r z \end{cases} \tag{3}$$

then move to show how this process can be reverse engineered such that an indicator may be designed to work within a specific regime of frequencies.

Since our ODEs are nonlinear many of the elegant results pertaining to frequency response of linear systems is unavailable. Instead, we must suppose a range of frequencies at which $s$ oscillates then note the impacts on our state variables of interest. Specifically, take

$$s(t) = A \sin(\omega t) + c$$

for parameters $A, \omega, c \in \mathbb{R}$. This process is showcased for a fixed value $\omega$ in Fig 10. The response in calcium ion $[\text{Ca}^{2+}] = x$ and indicator $[\text{CI}^*] = z$ follow from the supposed $s$ in the leftmost plot.

To avoid confusion, note $s(t)$ is the firing rate at given time $t$ and is measured in Hz. Here we examine how frequently $s$ must change before state variables begin to lag. This is

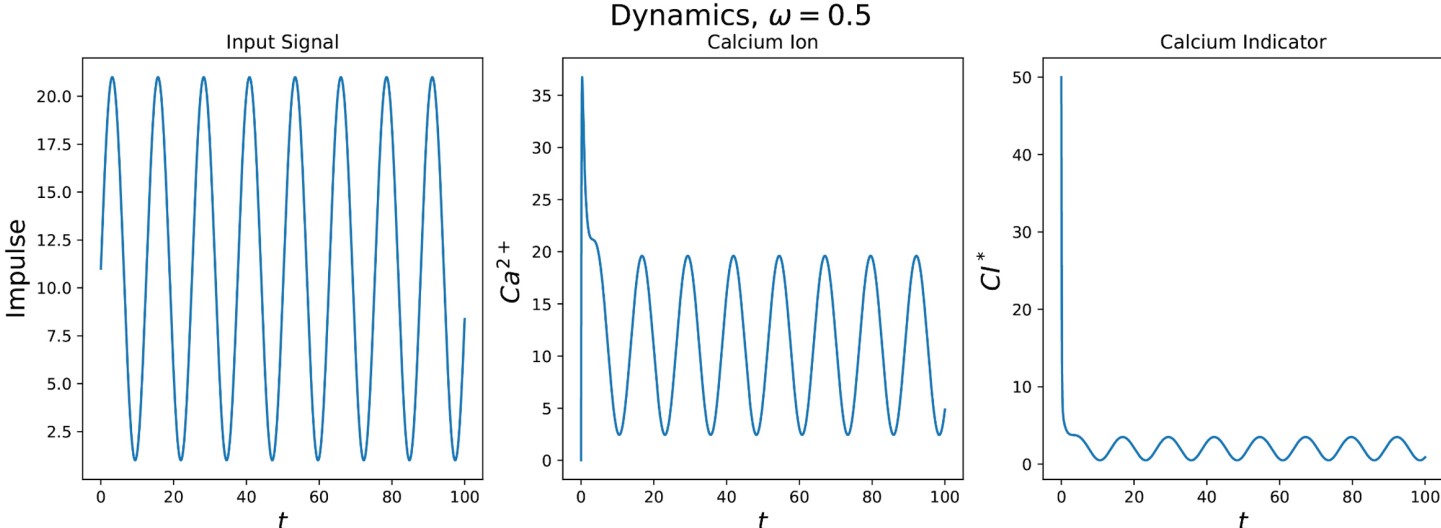

**Fig 10. Parameters:** $A = 20$, $c = 1$, $\omega = 0.5$. Notice after transient dynamics the resultant amplitudes for the calcium ion, indicator are not as large as the amplitude of the $s$ signal. This suggests these state variables do not keep up with changes in $s$ at this frequency.

determined by the parameter $\omega$, which determines the frequency of the $s$ signal, also measured in Hz. For the previous example, $10\sin(0.5t) + 11$ has period $4\pi$ and thus oscillates at a frequency of $1/(4\pi) \approx 0.080$ Hz between the values of 1 and 21 Hz.

To evaluate the ODE system's performance for a range of $\omega$, we perform simulations akin to those done in Fig 10. However after transient dynamics conclude, we note the ratio of amplitudes between $s$ and our state variables. This process is showcased in Fig 11. In this way we are able to tease out at which frequencies our state variables are able to reasonably respond to. The below calcium indicator keeps up with changes in firing rate poorly for higher frequency oscillations, highlighting the need to convert recorded calcium indicator quantities to firing rate for certain downstream analyses in which a neuron may often change the rate at which it fires.

This not only has the ability to analyze frequency response for existing indicators, but the potential to aid in the development of novel indicators. For example, if we desired an indicator which operated better for smaller values of $\omega$ than GCaMP6s, we might try setting new $k_f$ and $k_r$ then examine the resultant frequency response plot. To this end, consider Fig 12, which achieves precisely this by utilizing $k_f = 0.01$ and $k_r = 10$ as reaction rates for this toy example.

The task at hand then becomes creating an indicator with binding affinity $k_d = k_f/k_r = 0.01/10$. If additional control over indicator dynamics is desirable, each parameter in (3) may be tuned such that the frequency response is satisfactory. Constructing an indicator which agrees with all parameters in our governing ODE equations is likely difficult, but nonetheless this has the potential to serve as an approximate guide for the creation of future indicators.

As discussed in [29], general performance criteria to consider in the construction of GECIs include

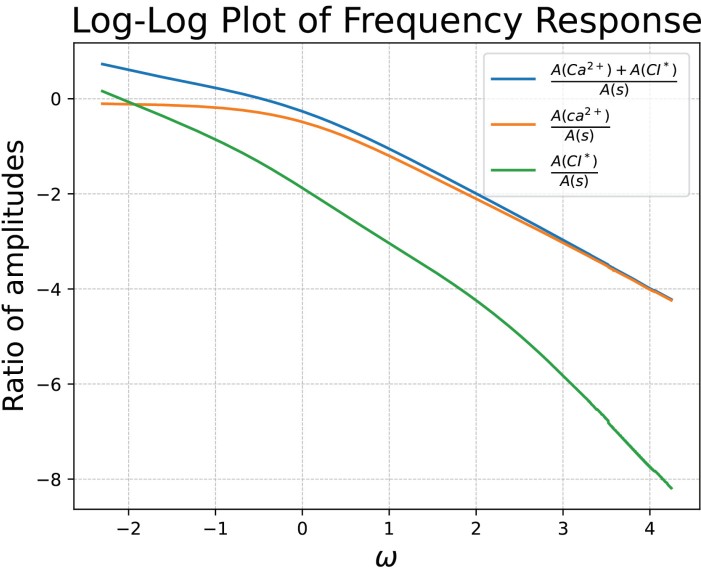

**Fig 11. Frequency response analysis of the commonly used indicator GCaMP6s, $k_f = 0.0514$, $k_r = 7.6$, $L = 30$.** Results show that for higher frequency oscillations in firing rate, calcium dynamics cannot keep up, evident in the amplitude of $s$ dominating the amplitude of calcium. Blue shows the total amount of calcium in the system, both bound and unbound. Orange depicts the unbound calcium ions themselves, while green shows the calcium ions bound to an indicator.

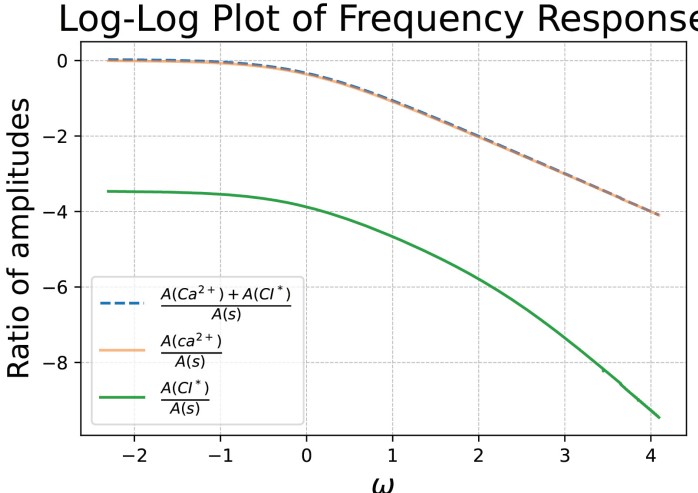

**Fig 12. Frequency response of a hypothetical indicator.** All parameters are held constant aside from $k_f$ = 0.01 and $k_r$ = 10. Notice an improvement in ratio of amplitudes at small $\omega$ when compared to Fig 11. A drop off still begins at $\omega$ = 0 or about $e^0$ = 1 Hz.

1. large dynamic range
2. high calcium sensitivity
3. faster response kinetics
4. linear response properties.

The frequency response analysis pipeline outlined in this section could prove valuable to evaluating the above performance metrics.

### Future work

This work presents a new method of detecting neuronal spikes in real-time from calcium imaging. The advantage of this method is the control theoretic approach that makes the algorithm suitable for real-time feedback control applications. Another advantage of this approach is the opportunity to improve the accuracy through further adaptations that can be explored.

As an avenue for improvement, consider the terminal penalty function of (4). This function can also be approximated via the Q function in reinforcement learning (RL) [30], presenting an opportunity for future directions of this work in applying RL-based approaches for efficient MPC design.

Next, a more sophisticated treatment of noise would increase accuracy further. This could be achieved by coupling another ODE connecting fluorescence to calcium, as is done by MLspike, or by employing particle filtering methods. Sequential Monte Carlo methods appear uniquely suited to address the stochasticity present in the system [31]. Earlier state of the art methods, like that from Vogelstein et al. [3], had success with this approach and has laid much of the necessary mathematical groundwork. Moreover, their auto-calibration approach avoids the need for simultaneous recordings of calcium and spiking activity, something our implementation currently relies on.

Finally, an in depth study of real time methods would further advance this work. To rigorously vet real time merit, a combination of in silico and in vivo experiments could be devised in which the accuracy of Oasis and MPC are demonstrated as a function of temporal lag.

Since MPC does not rely on backtracking, it would be worthwhile to explore if there exists some threshold for which MPC returns consistently more accurate results.

## Conclusion

In this paper, we propose a novel modeling framework and strategy to infer underlying firing rates of neurons provided calcium imaging traces. This first principles approach underscores the potential of adapting optimal control architectures to this space of problems. Accurate enough to challenge state of the art methods, this formulation joins the select few methodologies capable of providing real time spiking information. This methodology sets itself apart in that variable backtracking is not required. Leveraging a mechanistic model could lend to interpretability and can help in uncovering and predicting biophysical nuances as they relate to hyperparameters such as reaction rates of indicators.

## Supporting information

**S1 Appendix. Stability analysis of the ODE system** (3).
(PDF)

**S1 Appendix Fig A. Flows of equation** (3).
(PDF)

**S2 Appendix. Auto-calibration of parameters, in particular $\alpha$.**
(PDF)

**S1 Table. Parameter values and physical interpretation.**
(PDF)

## Acknowledgments

Thank you to the Gomez lab group, in particular Ksenia Zlobina, for insightful discussions pertaining to modeling and analysis.

## Author contributions

**Conceptualization:** Nicholas Rondoni, Fan Lu, Daniel B. Turner-Evans, Marcella Gomez.

**Data curation:** Nicholas Rondoni.

**Formal analysis:** Nicholas Rondoni.

**Funding acquisition:** Marcella Gomez.

**Investigation:** Nicholas Rondoni.

**Methodology:** Nicholas Rondoni, Fan Lu, Daniel B. Turner-Evans, Marcella Gomez.

**Project administration:** Daniel B. Turner-Evans, Marcella Gomez.

**Software:** Nicholas Rondoni.

**Supervision:** Daniel B. Turner-Evans, Marcella Gomez.

**Validation:** Nicholas Rondoni, Daniel B. Turner-Evans, Marcella Gomez.

**Visualization:** Nicholas Rondoni.

**Writing – original draft:** Nicholas Rondoni.

**Writing – review & editing:** Nicholas Rondoni, Fan Lu, Daniel B. Turner-Evans, Marcella Gomez.

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
