## [Decision Letter · Decision Letter 0]

5 Feb 2025

PCOMPBIOL-D-24-01903

Predicting neuronal firing from calcium imaging using a control theoretic approach

PLOS Computational Biology

Dear Dr. Rondoni,

Thank you for submitting your manuscript to PLOS Computational Biology. After careful consideration, we feel that it has merit but does not fully meet PLOS Computational Biology's publication criteria as it currently stands. Therefore, we invite you to submit a revised version of the manuscript that addresses the points raised during the review process.

In particular, although all the reviewers recognize the originality and potential importance of the approach presented, they raise similar issues. These include: 1) that the current accuracy obtained for this method falls below the state of the art, limiting the likelihood that it would be adopted by other researchers; 2) that auto-calibration to set parameters is also needed if the method is to be useful to the field; 3) that the variability in the obtained results is not sufficiently explored or explained. These problems are crucial to address in your revision.

Please submit your revised manuscript within 60 days Apr 07 2025 11:59PM. If you will need more time than this to complete your revisions, please reply to this message or contact the journal office at ploscompbiol@plos.org. Please include the following items when submitting your revised manuscript:

We look forward to receiving your revised manuscript.

Kind regards,

Barbara Webb

Academic Editor

PLOS Computational Biology

Joseph Ayers

Section Editor

PLOS Computational Biology

Feilim Mac Gabhann

Editor-in-Chief

PLOS Computational Biology

Jason Papin

Editor-in-Chief

PLOS Computational Biology

**Journal Requirements:**

3) Please ensure that all Figure and Table files have corresponding citations and legends within the manuscript. Currently, Figure 4, and Tables (1-3) in your submission file inventory do not have in-text citations. Please include the in-text citations of the figure and the tables.

4) We notice that your supplementary Tables, and information are included in the manuscript file. Please remove them and upload them with the file type 'Supporting Information'. Please ensure that each Supporting Information file has a legend listed in the manuscript after the references list.

**Reviewers' comments:**

Reviewer's Responses to Questions

**Comments to the Authors:**

**Please note that one of the reviews is provided as an attachment.**

Reviewer #1: In this manuscript, the authors proposed an approach to infer neuronal spiking times from calcium imaging signals. By employing modern control architectures, specifically model predictive control (MPC), within a chemical reaction network framework, the authors demonstrated accurate, fast, and interpretable results for predicting firing times. Furthermore, this framework is not limited to inference but also offers insights into biophysical nuances through the hyperparameters of calcium indicators. In total, the study presents a useful modeling framework for inferring neuronal firing rates from calcium imaging traces. However, the results remain preliminary, and further in-depth studies are necessary to justify publication in PLOS Computational Biology.

Major concerns:

1. Accuracy issue. The authors compared their MPC algorithm to two state-of-the-art algorithms, STM and Oasis, in Figure 2. While the authors highlighted that MPC offers better interpretability than Oasis and requires less training data than STM, its accuracy is inferior. Given that accuracy is a primary concern for users converting calcium signals into spike trains, the performance of MPC should be further improved to enhance its practical utility.

2. A significant limitation of the MPC approach is its sensitivity to the chosen dataset. As seen in Figures 3 and 4, the correlation scores of MPC are inconsistent, with median values ranging from below 0.2 to nearly 0.6 depending on the dataset. Furthermore, the MPC method shows greater variability in its mean correlation scores compared to STM. This instability may lead to doubts about the reliability of results obtained using MPC.

3. The current implementation of the MPC model lacks robust auto-calibration functionality. Manual parameter adjustments are inefficient and require substantial biophysical knowledge from the user, making the model less user-friendly. Considering the noisy nature of calcium signaling data, parameter selection becomes even more challenging. Figure 7 highlights that the model's noise-resistance capabilities need further optimization to enhance usability.

4. While the authors emphasized the availability of the MPC model for online control, Figure 6 showed temporal deviations between Ssim and Smeas, in particular for the first spike in the 10-second subset. If such deviations persist during online control, they could destabilize the closed-loop system. The authors should address whether these deviations impact real-time applications.

5. The frequency response analysis pipeline described in the final section is undoubtedly valuable for understanding biological processes. However, it remains unclear how these biological insights translate into improved model performance. While the chemical reaction-based modeling framework has strong biological interpretability, further optimizations are needed to enhance both the performance and practicality of the method.

Minor concerns:

1. The full name of NND is missing from the text. Is it Non-Negative Deconvolution?

2. In line 253, the word “biophysical” is misspelled as “biophyiscal.”

Reviewer #2: The manuscript introduces a novel method to predict neuronal firing rates from single cell calcium imaging data. The algorithm uses a mechanistic model based on chemical reaction networks (CRN) and model predictive control (MPC). This approach offers real-time computation capabilities and higher interpretability respect to the state of the art, especially as it is based on a model of calcium dynamics, addressing a key limitation of existing methods that rely heavily on neural networks or non-mechanistic strategies.

Major points:

As the topic of the study is the inference of spiking activity from calcium imaging traces, and since the vast majority of learning-based methods have been shown to be still outperformed by the much more simple nonnegative deconvolution (NND) 1, the authors correctly compare their method with NND.

The authors show that their method is outperformed by both the STM and NDD methods. While there is an argument that the efficiency of STM methods varies greatly depending on their training, this is not true for NND.

All in all, the method proposed by the authors offers improved interpretability at the cost of speed and accuracy, respect to NND. However, while I do find interpretability to be an extremely important factor in developing new methodologies, the target audience for this method is likely to care more about its predictive ability, and, secondarily, speed.

Minor points:

- I find it unclear why the authors refer specifically to two-photon calcium imaging, and not to calcium imaging in general. What is necessary to apply their analysis is to have calcium imaging at the single-cell resolution, not necessarily with a two-photon microscope. I see no reason why their analysis should not apply to 1-photon or multi-photon calcium imaging. Given the expertise of the authors with applied mathematics, I believe this might be since they have mostly cooperated with a laboratory that focuses on two-photon. I recommend the authors to adjust the manuscript accordingly, since their method is more widely applicable.

- The requirement of having to tune the parameters for optimal performance, while being positive as it introduces an extra degree of control over the method, is similarly likely to constitute a potential source of error for least experienced experimenters.

- Line 37, the authors explain for the second time the meaning of MPC, this is unnecessary as it has been explained already in line 28. I believe that instead, the STM acronym has not been explained.

- Figure 2, what is NDD* in the legend?

- Since the method seems very promising in taking into account different calcium indicator dynamics, it might be worth adding a comparison of performance with other methods in case of different calcium indicators (e.g. GCaMP6s vs GCaMP6f, or even better, RCaMP). It might potentially greatly improve the applicability of the method, since standard methods tend to struggle with RCaMP.

References

Pachitariu, Marius, Carsen Stringer, and Kenneth D. Harris. "Robustness of spike deconvolution for neuronal calcium imaging." Journal of Neuroscience 38.37 (2018): 7976-7985.

Reviewer #3: I assume all the code will be public, and the results fully reproducible, as a pre-condition for acceptance in PLoS CB. However, I did not check either of those things.

Please also see attached pdf for reviewer #3

**Have the authors made all data and (if applicable) computational code underlying the findings in their manuscript fully available?**

Reviewer #1: **No: **

Reviewer #2: Yes

Reviewer #3: None

PLOS authors have the option to publish the peer review history of their article (what does this mean?). If published, this will include your full peer review and any attached files.

Reviewer #1: No

Reviewer #2: **Yes: **Niccolo' Calcini

Reviewer #3: **Yes: **Joshua T. Vogelstein

**Figure resubmission:**
---

## [Decision Letter · Decision Letter 1]

19 May 2025

Dear Mr. Rondoni,

We are pleased to inform you that your manuscript 'Predicting neuronal firing from calcium imaging using a control theoretic approach' has been provisionally accepted for publication in PLOS Computational Biology.

Best regards,

Barbara Webb

Academic Editor

PLOS Computational Biology

Hugues Berry

Section Editor

PLOS Computational Biology

Reviewer's Responses to Questions

**Comments to the Authors:**

Reviewer #1: The author answered my question well in the revised version. I have no more comment now.

Reviewer #2: I thank the authors for having addressed all my points and doubts to the best of their ability. I find that the addition of the auto-calibration pipeline, especially if user-friendly enough, would be an excellent improvement of the method, and allow more researchers to access it, especially as the field still has a strong split between experimenters and analysis/math-savy experts.

While indeed the method is either roughly comparable with or outperformed by previous methods (STM and Oasis respectively), the method, with respect to these, allows for real time feedback control, which is a feature that might be of interest for other groups and makes it of potential interest for publication. The authors also reported that the calibration slightly improved the method’s score, bringing it fully on par with STM.

Thanks to this, and the reactions to my other minor points and the points made by the other reviewers, I believe the manuscript has improved and been more solid, and the method more usable and accessible.

**Have the authors made all data and (if applicable) computational code underlying the findings in their manuscript fully available?**

Reviewer #1: None

Reviewer #2: Yes

PLOS authors have the option to publish the peer review history of their article (what does this mean?). If published, this will include your full peer review and any attached files.

Reviewer #1: **Yes: **Jiulin Du

Reviewer #2: **Yes: **Niccolò Calcini

---

## [Editor Report · Acceptance letter]

PCOMPBIOL-D-24-01903R1

Predicting neuronal firing from calcium imaging using a control theoretic approach

Dear Dr Rondoni,

I am pleased to inform you that your manuscript has been formally accepted for publication in PLOS Computational Biology. Your manuscript is now with our production department and you will be notified of the publication date in due course.

With kind regards,

Zsofia Freund
